

# Seasonal stratospheric ozone trends over 2000-2018 derived from several merged data sets.

Monika E. Szeląg[1], Viktoria F. Sofieva[1], Doug Degenstein[2], Chris Roth[2], Sean Davis[3] and Lucien Froidevaux[4]

[1] Finnish Meteorological Institute, Helsinki, Finland
[2] University of Saskatchewan, Canada
[3] NOAA Earth System Research Laboratory Chemical Sciences Division, Boulder, CO, USA
[4] Jet Propulsion Laboratory, California Institute of Technology, Pasadena, California, USA

*Correspondence to*: Monika Szelag (monika.szelag@fmi.fi)

## Abstract

In this work, we analyse the seasonal dependence of ozone trends in the stratosphere using four long-term merged datasets: SAGE-CCI-OMPS, SAGE-OSIRIS-OMPS, GOZCARDS and SWOOSH which provide more than 30 years of monthly zonal mean ozone profiles in the stratosphere. We focus here on trends between 2000 and 2018. All datasets show similar results,

although some discrepancies are observed. In the upper stratosphere, the trends are positive throughout all seasons and the majority of latitudes. The largest upper stratospheric ozone trends are observed during local winter (up to 6% dec$^{-1}$) and equinox (up to 3% dec$^{-1}$) at mid-latitudes. In the equatorial region, we find a very strong seasonal dependence of ozone trends at all altitudes: the trends vary from positive to negative, with the sign of transition depending on altitude and season. The trends are negative in the upper stratospheric winter (-1 to -2% dec$^{-1}$) and in the lower stratospheric spring (-2 to -4% dec$^{-1}$), but positive

(2-3% dec$^{-1}$) at 30-35 km in spring, while the opposite pattern is observed in summer. The tropical trends below 25 km are negative and maximize during summer (up to -2 % dec$^{-1}$) and spring (up to -3% dec$^{-1}$). In the lower mid-latitude stratosphere, our analysis indicates hemispheric asymmetry: during local summers and equinoxes, positive trends are observed in the South (+1 to +2% dec$^{-1}$) while negative trends are observed in the North (-1 to -2% dec$^{-1}$).

We compare the seasonal dependence of ozone trends with available analyses of the seasonal dependence of stratospheric

temperature trends. We find that ozone and temperature trends show positive correlation in the dynamically controlled lower stratosphere, and negative correlation above 30 km, where photochemistry dominates.

Seasonal trend analysis gives information beyond that contained in annual mean trends, which can be helpful in order to better understand the role of dynamical variability in short-term trends and future ozone recovery predictions.

## 1    Introduction

The stratospheric ozone layer plays an important role in the energy budget and dynamics of the middle atmosphere by absorbing a great part of harmful ultraviolet solar radiation. Its changes in the stratosphere contribute to ground-level climate variability. International efforts by scientists and politicians resulted in the Montreal Protocol agreement and its amendments, which regulate ozone-depleting substances (ODSs). Upper stratospheric ozone declined by about 4-8 % dec$^{-1}$ in the upper

stratosphere from 1980 to the late 1990s (Steinbrecht et al., 2017a; WMO, 2018). The Antarctic ozone hole is showing some signs of recovery, and the first signatures of global recovery have now been observed in the upper stratosphere (Bourassa et



al., 2014; Kyrölä et al., 2013; Newchurch et al., 2003; Tummon et al., 2015; WMO, 2018). However, no significant trend has been detected in global total column ozone and even though upper stratospheric ozone is recovering, negative trends have been reported in the lower stratosphere (Ball et al., 2018; WMO, 2018). Because of the strong linkage and feedbacks between ozone depletion and climate change, continuous monitoring of ozone is very important for understanding its variability and its role in climate change. Furthermore, stratospheric ozone depletion impacts stratospheric temperature, which is also a key indicator of global climate variability. The majority of ozone trend studies have assumed that the trends can be described as a simple piecewise-linear function (Bourassa et al., 2014; Kyrölä et al., 2013; Nair et al., 2015; Neu et al., 2014; Sofieva et al., 2017; Steinbrecht et al., 2017). On the other hand, more comprehensive analyses of stratospheric temperature trends have revealed large seasonal variability (Funatsu et al., 2016; Khaykin et al., 2017; Randel et al., 2016). Funatsu et al. (2016) reported cooling in the stratosphere over the period 2002-2014, at a rate of about 0.5 K dec$^{-1}$ above 25 km. They have also shown large seasonal variability at midlatitudes, with significant negative trends of -0.6 to -1 Kdec$^{-1}$ during summer and autumn. Khaykin et al. (2017) revealed a middle stratospheric cooling over 2002–2016 at an average rate of -0.1 to -0.3 K dec$^{-1}$, with seasonal trend patterns indicating changes in the stratospheric circulation. Randel et al. (2016) have pointed to a stratospheric cooling varying from -0.1 to -0.6 K dec$^{-1}$ over 1979-2015, with larger cooling during 1979–1997 compared to 1998–2015, mirroring the differences in upper-stratospheric ozone trends.

This paper is dedicated to an investigation of the seasonal dependence of ozone trends in the stratosphere for the post-2000 time period. For our analysis, we have used several merged satellite datasets that cover more than three decades (from 1984 to 2018). Ozone trends are estimated based on two-step multiple linear regression. The paper is organized as follows. Section 2 describes the merged ozone datasets. Section 3 describes the methods used for seasonal trends estimation. Section 4 describes our results regarding the seasonal dependence of ozone trends in the stratosphere. Our conclusions are summarized in Section 5.

## 2    Data

In order to analyze seasonal stratospheric ozone trends, four long-term merged data sets of ozone profiles have been used.

These datasets are:

- SAGE-CCI-OMPS dataset developed in the framework of ESA Climate Change Initiative (Sofieva et al., 2017), referred to as CCI hereafter
- SAGE-OSIRIS-OMPS developed at the University of Saskatchewan (Bourassa et al., 2014), referred to as SOO hereafter
- The Global Ozone Chemistry And Related trace gas Data records for the Stratosphere (GOZCARDS) (Froidevaux et al., 2015, 2019)
- The Stratospheric Water and Ozone Satellite Homogenized (SWOOSH) created by NOAA/ESRL Davis et al. (2016).

General information about the merged datasets is collected in Table 1. The basic information about individual ozone profile datasets used in the merged datasets is collected in Table 2. The collection of merged datasets represents two main groups categorized according to ozone profile representation: number density on altitude grid (CCI and SOO) and volume mixing ratio (vmr) on pressure grid (GOZCARDS and SWOOSH). The vmr-based group uses MLS/Aura as the main dataset after August 2004, while the number-density group uses other satellite instruments, with retrievals on altitude grid, after 2005. The merged datasets use different collections of limb-profiling instruments and use different merging methods (see Table 1 and references for more details). All merged datasets are constructed from datasets having a good vertical resolution of 2-4 km.




More detailed information about the merged datasets can be found in the publications mentioned above and in Table 1. All the merged datasets have been extended until December 2018. These long-term monthly zonal mean datasets cover all seasons.

**Table 1. General information about the merged datasets**

| Dataset | SAGE-CCI-OMPS | SAGE-OSIRIS-OMPS | GOZCARDS (v2.20) | SWOOSH (v.2.6) |
|---|---|---|---|---|
| **Time coverage** | Oct 1984- Dec 2018 | Oct 1984- Dec 2018 | Jan 1979-Dec 2018 | Oct 1984- Dec 2018 |
| **Latitude coverage and representation** | 90°S-90°N 10° zones | 60°S-60°N 10° zones | 90°S-90°N 10° zones | 90°S-90°N 10° zones (also 5°, 2.5 °) |
| **Vertical coverage and sampling** | 10-50 km, 1-km grid | 10-50 km, 1-km grid | 215-0.2 hPa 12 levels dec$^{-1}$ (~ 3 km) | 316- 1 hPa 12 levels dec$^{-1}$ (~ 3 km) |
| **Included instruments** | SAGE II OSIRIS OMPS-LP GOMOS MIPAS SCIAMACHY ACE-FTS | SAGE II OSIRIS OMPS-LP | SAGE I SAGE II HALOE Aura MLS | SAGE II HALOE UARS MLS Aura MLS SAGE III |
| **Merging method** | using deseasonalized anomalies and the median | using deseasonalized anomalies, SAGE II is reference | Offsets to SAGE II (reference) and averaging | Offsets to SAGE II (reference) and averaging |
| **Ozone profile representation** | deseasonalized anomalies and ozone concentrations | deseasonalized anomalies | vmr | vmr |
| **Reference** | Sofieva et al. (2017) | Bourassa et al. (2014), Zawada et al. (2017) | Froidevaux et al., (2015, 2019) | Davis et al. (2016) |

5    **Table 2 Basic information about individual satellite datasets used in the merged datasets**

| Instrument/platform | Acronym explanation | Used period | Data processor |
|---|---|---|---|
| SAGE I/AEM-B | Stratospheric Aerosol and Gases Experiment I | 1979-1981 | V5.9 rev |
| SAGE II/ERBS | Stratospheric Aerosol and Gases Experiment II | 1984-2005 | V7 |
| SAGE III/Meteor-3M | Stratospheric Aerosol and Gases Experiment III | 2002-2005 | V4 |
| HALOE/UARS | HALogen Occultation Experiment | 1991-2005 | V19 |
| MLS/UARS | Microwave Limb Sounder | 1991-1997 | V5 |
| MLS/Aura | Microwave Limb Sounder | 2004-2018 | V4.2 |
| OSIRIS/Odin | Optical Spectrograph and InfraRed Imaging System | 2001-2018 | v5.10 |
| MIPAS/Envisat | Michelson Interferometer for Passive Atmospheric Sounding/ | 2005-2012 | IMK/IAA v7 |
| SCIAMACHY/Envisat | SCanning Imaging Absorption spectroMeter for Atmospheric ChartographY | 2003-2012 | UB v3.5 |
| GOMOS/Envisat | Global Ozone Monitoring by Occultation of Stars | 2002-2011 | ALGOM2s v1 |
| ACE-FTS/SCISAT | Atmospheric Chemistry Experiment Fourier Transform Spectrometer | 2004-2018 | v3.5/3.6 |
| OMPS-LP/Suomi-NPP | Ozone Monitor Profiling Suite-Limb Profiler | 2012-2018 | Usask2D v1.1.0 |

**Methods**


The seasonal trend analysis was performed on monthly deseasonalized anomalies for all merged data sets. The CCI and SOO datasets provide deseasonalized ozone anomalies. For GOZCARDS and SWOOSH, the deseasonalized anomalies were computed in the same way. The seasonal cycle was evaluated using the data from 2005-2011.

The trend analyses are usually performed using multiple linear regression technique, in order to separate the natural variability and long-term trends. The known main sources of ozone variability, which are characterized by corresponding proxies, are solar cycle, QBO and ENSO. In many studies, the regression is performed in one step by assuming an approximation of ozone trends by a piecewise linear function ( e.g., (Bourassa et al., 2014; Harris et al., 2015; Kyrölä et al., 2013; Sofieva et al., 2017)). Alternatively, the regression can be performed in two steps, with detecting natural cycles and removal them on the first step and estimating bulk changes in some periods in the second step (e.g., (Steinbrecht et al., 2017; WMO, 2018)). The two-step approach avoids fitting near turnaround point, and thus it has reduced sensitivity to its choice. The two-step approach is equivalent to the one-step regression, in which piece-wise linear function has three segments, as used in the SPARC LOTUS report (Petropavlovskikh et al., 2019), independent-linear-trend (ILT) fitting method. The trend analysis can be also performed using the dynamical linear modelling (e.g.(Ball et al., 2019; Laine et al., 2014)), which assumes that long-term changes are described by a smooth function in time.

Analyses of the seasonal dependence of trends are based on smaller number of data points, compared to the traditional trend analyses, in which data from all months are used. For example, if one analyzes trends in a given month, a 30-yr long dataset will have only 30 data points. The fitting of all proxies in the standard multiple linear regression with insufficient data points will result in substantial uncertainty of estimated parameters. Therefore, we propose the following two-step multiple regression for seasonal trend analysis. In the first step, natural cycles are estimated from the data using the traditional multiple linear regression:

$$O_3(t) = PWLT(t, t_0) + q_1 QBO_{30}(t)) + q_2 QBO_{50}(t)) + sF_{10.7}(t)) + dENSO(t), \qquad (1)$$

where ozone trends are approximated with a piecewise linear function ($PWLT(t, t_0)$) with the turnaround point in 1997, $QBO_{30}(t_s)$ and $QBO_{50}(t_s)$ are equatorial winds at 30 and 50 hPa, respectively (http://www.cpc.ncep.noaa.gov/data/indices/), $F_{10.7}(t_s)$ is the monthly average 10.7 cm solar radio flux (ftp://ftp.geolab.nrcan.gc.ca/data/solar_flux/monthly_averages/), and $ENSO(t_s)$ is the 2-month lagged ENSO proxy (http://www.esrl.noaa.gov/psd/enso/mei/table. html). Then the fitted natural cycles are removed from the data, so only smooth variations remain in the resulting time series. The regression Eq. (1) can be performed using the data only from a certain period, usually 3 months (referred to as method #1 hereafter) or using all available data points (we will call this method #2). In the second step, the trends are estimated separately for the decline (1984-1997) and the recovery period (2000-2018) using a simple linear regression. Autocorrelations are removed using the Cochrane–Orcutt transformation (Cochrane and Orcutt, 1949). In the two-step approach, sufficient number of data points are available for detection of natural cycles - solar, QBO and ENSO, thus providing more accurate fitting of these proxies. In the second step, the fitting is not performed near the trend turnaround point in 1997, thus providing smaller sensitivity to defining turnaround point, compared to the one-step multiple linear regression with the "hockey-stick" trend term.

Figure 1 illustrates two steps of our analysis for the CCI dataset and for two selected periods (MAM, SON) in the tropics (10°S-10°N) at altitudes between 30-35 km. Data from seasons (3-months, method #1) are used in the first step (Eq.(1)). The original ozone anomalies (top panel of Figure 1) show a variability of about ±10% during 1984-2018 for both seasons. The anomalies are reduced to about ±6% after the first step of the regression analysis where all proxies ($QBO_{30}$, $QBO_{50}$, $F_{10.7}$, $ENSO$) are subtracted from the original data (middle panel of Figure 1). Finally, the bottom panel of Figure 1 illustrates the


second step of the regression analysis, i.e., linear trends are estimated for the 2000-2018 in April (trend of +2.3% dec[-1]) and October (trend of -2.1% dec[-1]).

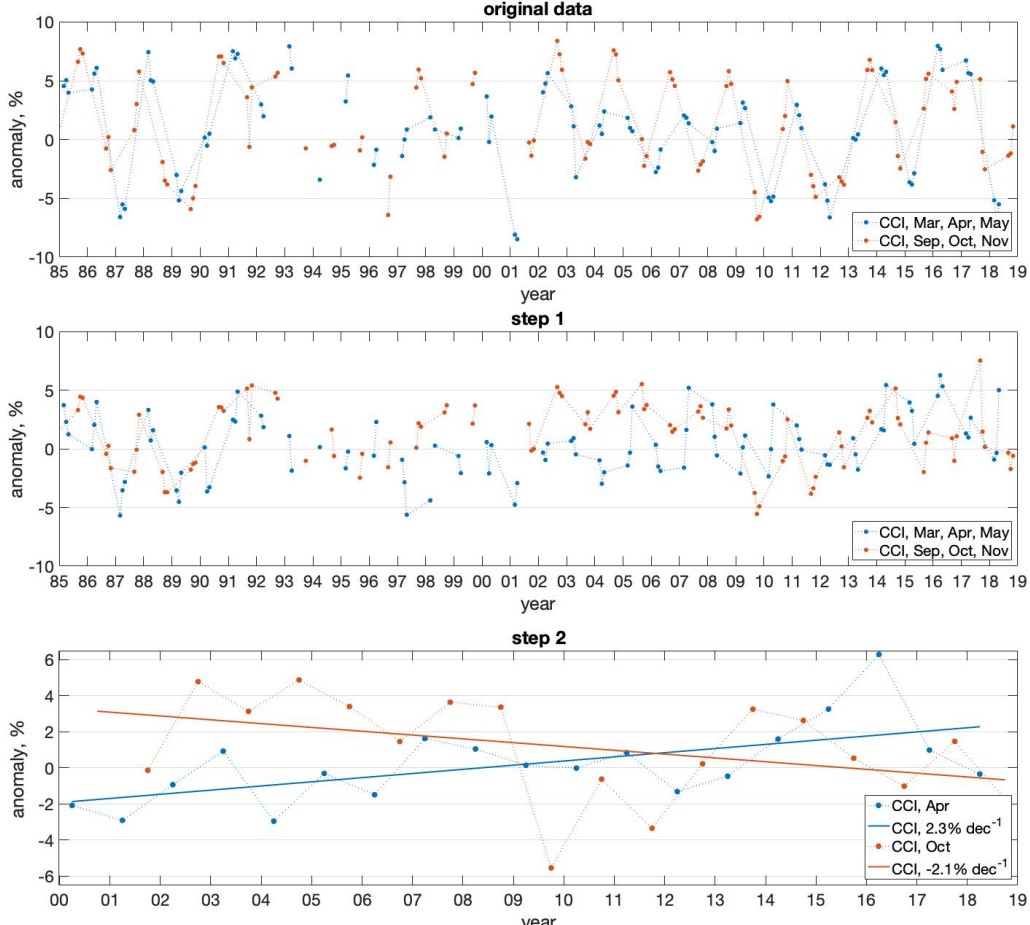

**Figure 1. Two-step multiple linear regression example from CCI for 10°S-10°N and 30-35 km. Top panel: original ozone anomalies.**
**Middle panel: anomalies with cycles removed. Bottom panel: linear trends estimated for two selected months.**

As described above, the amplitude of the natural cycles (solar, QBO, ENSO), which will be removed from the time series, can be estimated in two ways in the first step of our regression method, i.e., for each season separately (method #1) or using the data from all months, like in the traditional trend analysis (method #2). We have investigated both methods for estimating the
natural cycles and we found that in general, uncertainties of method #1 are smaller. This is illustrated in Figure 2, which shows the ozone trends at latitudinal bands 10°S-10°N (top panels) and 30°N-60°N (bottom panels) in April for these two methods (left and middle panels). The right panel of Figure 2 shows the differences in trends and uncertainties between both methods. At mid-latitudes, the difference in uncertainties is negligible (less than 0.3%), but method#1 provides some smaller trend uncertainties, especially around ~35 km.

Such an observation is counter-intuitive: one would expect better fitting using more data points. The reason for this might be the seasonal dependence of natural cycles, particularly the QBO (Gabis et al., 2018). Assuming this, one would expect larger

differences between the two methods in the tropical region, where the QBO dominates, and this is observed in Figure 2. Although our observations – smaller residuals when fitting natural cycles using the data from a 3-months season – seem to support the hypothesis of seasonal dependence of natural cycles, this discussion is beyond the scope of our paper.

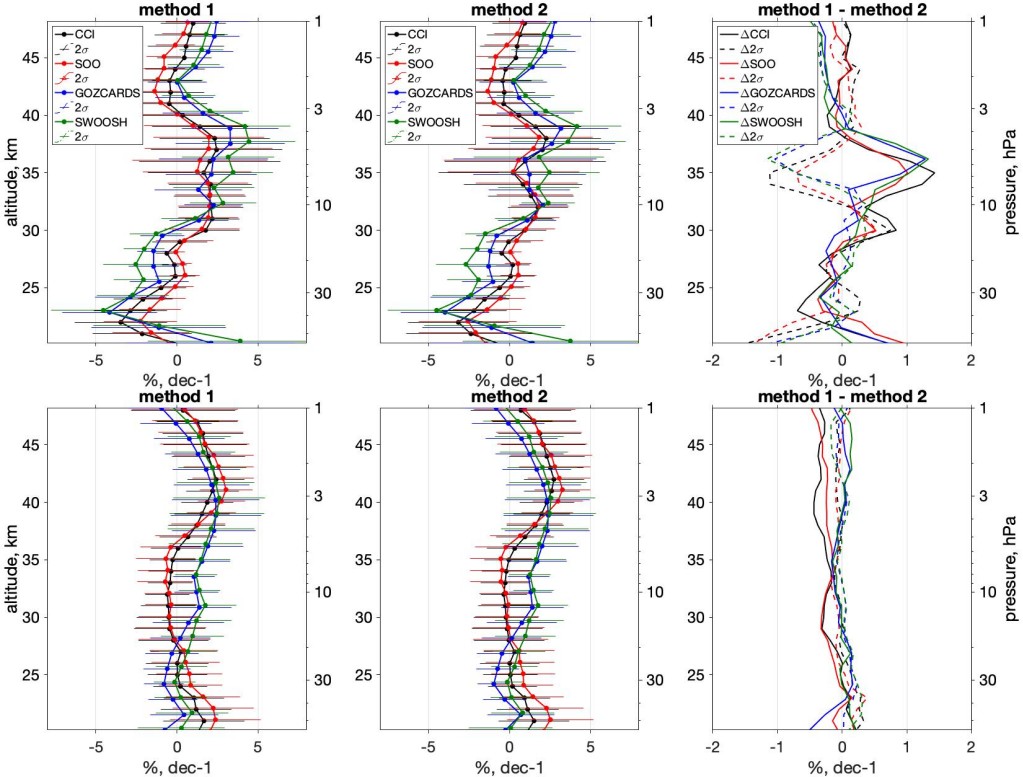

**Figure 2. Vertical profiles of ozone trends in 2000-2018 from CCI (black), SOO (red), GOZCARDS (blue) and SWOOSH (green). The results are shown for April, latitudinal bands 10°S-10°N (top panels) and 30°N-60°N (bottom panels) for two separate methods (see description in the text). Error bars are 2-sigma uncertainties. Data are presented on their natural vertical coordinate: altitude grid (left axis) for CCI and SOO and pressure grid (right axis) for GOZCARDS and SWOOSH. Left panel shows the difference between methods for trends (solid lines) and uncertainties (dashed lines).**

In the analyses shown below, we have used method #1 for the evaluation and removal of natural cycles. We would like to emphasize, that the results are similar if method #2 is used (see text below and Supplementary information for details). In this work, we will focus and discuss post-2000 trends only.

## 4 Results

The seasonal variation of ozone trends over the 2000–2018 period as a function of latitude for 5 selected altitude regions is shown in Figure 3 (colored contours). The red/blue shading (positive/negative trends) in Figure 2 denote trends that are statistically significant at the 95% level.





In the tropical region (20°S-20°N), a strong seasonal dependence of ozone trends is observed. In the lower stratosphere (19-23 km), the pattern of statistically significant negative trends of about -2 to -3% dec⁻¹ is present in all merged data sets during the spring and summer months (MAM and JJA), while they are less pronounced in other months. At altitudes 24-28 km, the trends change from negative (-1 to -2% dec⁻¹, statistically significant for all except SOO) in spring (MAM) to positive (+1 to +2% dec⁻¹, statistically significant for CCI and SOO) in autumn (SON). At 31-35 km in the tropics, the trends are opposite: positive trends in MAM (+2 to +3% dec⁻¹, statistically significant) and negative in SON (-1 to -2% dec⁻¹, not statistically significant).

Above 40 km, trends are in general positive throughout the latitudes and months, with the largest trends observed at mid-latitudes (30°N/S-60°N/S) during the local winters and spring/autumn seasons (2-4% dec⁻¹). Upper stratospheric recovery for all latitude bands has been obtained by others (e.g., Petropavlovskikh et al., 2019; Steinbrecht et al., 2017; WMO, 2018), nonetheless, for seasonal analysis, negative trends of -1 to -2% dec⁻¹ during the late winter (DJF) are present in the tropics, although with statistical significance only in the CCI datasets.

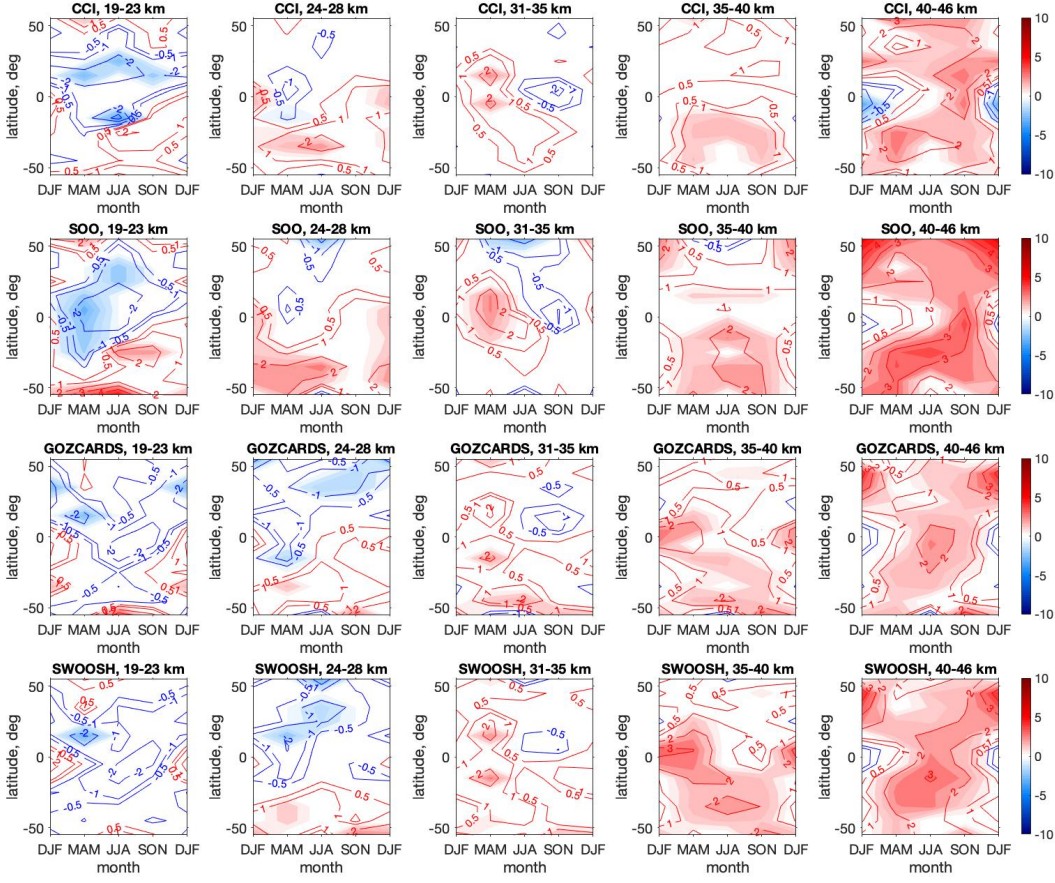

**Figure 3. Latitude-seasons variation of linear trends in ozone for each of the merged data sets calculated over 2000-2018 for five selected altitude/pressure bands. The shading denotes trends that are significant at the 95% level.**





Analyses performed for broader latitude bands, presented in Figure 4, reveal negative trends ranging from -1 to -2% dec$^{-1}$ in the lower/middle stratosphere at 30°N-60°N during the summer (JJA). Negative trends are present in all datasets and are statistically significant for all except the CCI dataset. In contrast, in the lower/middle southern stratosphere (30°S-60°S), trends are positive (1-2% dec$^{-1}$) and statistically significant for all datasets.

In the equatorial region, all datasets show pronounced, statistically significant and very similar seasonal dependence of ozone trends.  In the upper tropical stratosphere above 40 km, trends are negative in DJF (-1 to -3% dec$^{-1}$, significant for all except SOO), and positive in August-October (2-3 %, statistically significant for all datasets).  At 30-35 km, trends are positive in MAM at altitudes 30-35 km (2 to 3 % dec$^{-1}$, significant for all) and negative in SON (~ -1 % dec$^{-1}$, statistically significant for CCI and SOO).  In the lower stratosphere, the negative trends are in MAM (-2 to -3% dec$^{-1}$, significant for all) and positive in
DJF (1 to 2% dec$^{-1}$, significant for CCI and SOO).

Upper stratospheric trends at mid-latitudes are most pronounced during the local winters and equinoxes, varying from 3 to 4% dec$^{-1}$ in the North and from 2 to 3% dec$^{-1}$ in the South.

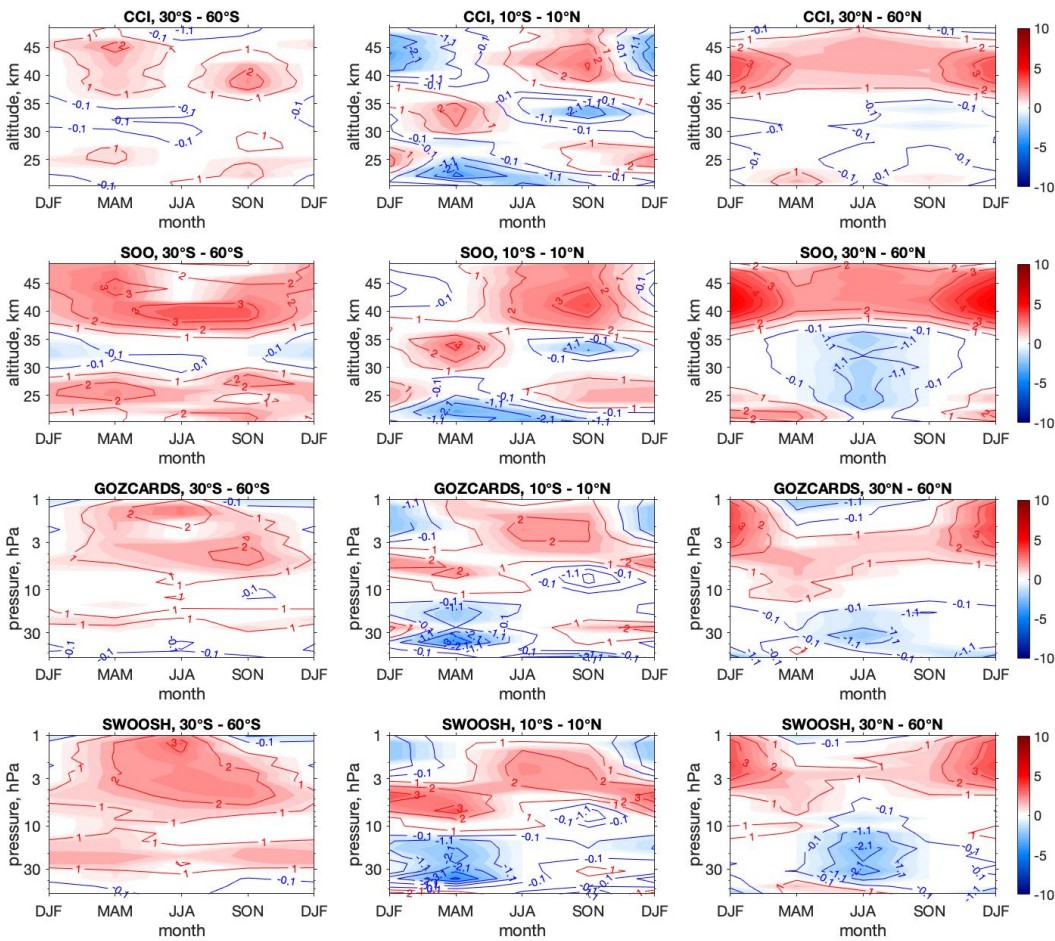

Figure 4. Altitude-seasons variation of linear trends in ozone for each of the merged data sets calculated over 2000-2018 for three
selected latitudinal bands. Data are presented on their natural vertical coordinate: altitude grid for CCI and SOO and pressure
grid for GOZCARDS and SWOOSH.  The shading denotes trends that are significant at the 95% level.




The basic structure and patterns of seasonal trends are apparent in the monthly data as well (Supplementary Figure S2 and Figure S3) but the magnitude of calculated trend is higher for each single month.

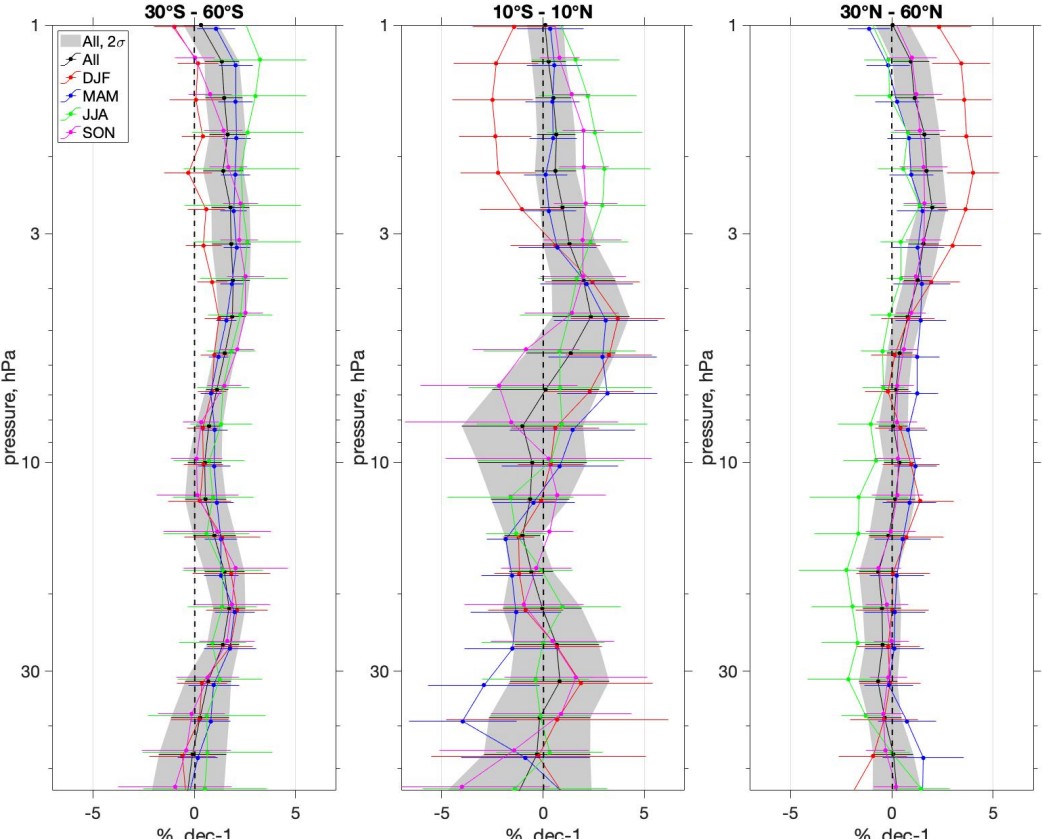

**Figure 5. Vertical profiles of seasonal (red, blue, green, magenta) and yearly (black) ozone trends in 2000-2018 from SWOOSH. The results are shown for three selected latitude bands. Error bars and shaded area (gray) are 2-sigma uncertainties.**

Figure 5 shows the seasonal ozone trends (color) together with yearly trend (black) plotted in the vertical distribution for three selected latitudinal bands. It is clear that at mid-latitudes, positive trends in the upper stratosphere are dominating over the local winters (up to 4% dec[-1]) and are much higher than the yearly trend (up to 2% dec[-1]). In the tropics, the main features
observed during different seasons (i.e. negative-positive patterns observed throughout the seasons and vertical levels) mostly cancel out in the yearly trend. Negative winter trends (DJF) in the upper stratosphere (-3% dec[-1]), negative spring trends (MAM) in the lower stratosphere (-3 to -4% dec[-1]), and positive spring trends (MAM) in the middle stratosphere (2% dec[-1]) are all counterbalanced by opposite or smaller trends during the remaining seasons. As a result, the yearly ozone trend in the tropics is much smaller than the seasonal trends. The other datasets show consistently similar features (Supplementary Figure
S3). The main discrepancies are observed between number density- based group (SOO and CCI) and vmr-based group (GOZCARDS and SWOOSH) during summer (JJA) in the upper stratosphere in the South.



## 5    Discussion

To summarize our analysis, variations of ozone trends over the period 2000–2018 for each latitude and vertical level are plotted for each season separately in Figure 6 (method #1) and Supplementary Figure S4 (method #2). Both methods as well as all datasets show, in general, very similar results. The uncertainties of method #1 are usually smaller than for method #2, which
makes the observed patterns more statistically significant.

In the upper stratosphere, trends are positive throughout all seasons and the majority of latitudes. One of the most pronounced features is that the mid-latitude upper stratospheric ozone trends are larger in local winters. As discussed in Sect 1., ozone and temperature trends are inter-related, as ozone and temperature are connected via photochemical reactions (effective above ~25-30 km, (Brasseur and Solomon, 2005). Randel et al. (2016) reported weaker upper-stratospheric cooling in local winter at mid-
and high latitudes. This is fully consistent with our observations of larger ozone trends at these locations in winter. A hypothesized explanation of this feature might be the acceleration of the upper branch of the Brewer-Dobson circulation (or in general mean residual circulation), which controls the meridional transport of trace gases from the tropical region to the poles (Brewer, 1949; Dobson, 1956). The Brewer-Dobson circulation is most effective during winter season (Chipperfield, M. P., Jones, 1999). Several modelling studies have shown that due to the greenhouse gas concentration (GHG) increases, the
winter time BDC will strengthen and accelerate the expected ozone recovery (Butchart et al., 2006; Garcia and Randel, 2008; Gettelman et al., 2010; Li et al., 2008; Schnadt et al., 2002; Sigmond et al., 2004). Also, observational studies have shown an accelerated BDC over the tropical region (Thompson and Solomon, 2009) as well as at high latitudes (Hu and Fu, 2009). Increased speed of the BDC would have an effect on the transport of ozone and ODSs. While the main reason for positive ozone trends in the upper mid-latitude stratosphere is due to the decrease of ozone-depleting substances, the seasonal variations
of the ozone trends can be due to dynamics. Seasonal dependence of both temperature and ozone trends support this hypothesis. However, the investigation of the mechanisms that controls the seasonality of ozone and temperature trends is beyond the scope of our paper; it can be the subject of future modelling and observational studies.

In the tropics, our analysis has shown a very strong seasonal dependence of ozone trends observed at all altitudes. The trends
change from positive to negative, with the phase changing with altitude. In the tropical lower stratosphere below 25 km, strong negative trends are observed during boreal spring and summer, which are statistically significant for all datasets. Khaykin et al., (2017) also found an altitude-dependent pattern in temperature trends in the tropical region, with a strong seasonality. One can notice that the changes of phases in ozone and temperature trends are very similar (temperature trends are evaluated below 35 km in Khaykin et al., (2017)). In the lower tropical stratosphere, in the dynamically controlled region, ozone and
temperature variations are positively correlated (Hauchecorne et al., 2010), so do the ozone and temperature trends (compare our Fig. 8 with Fig. 5 in Khaykin et al., 2017). This is rather expected and can serve as additional confirmation of hypotheses explaining this seasonality. Khaykin et al. (2017) hypothesized that the observed trends structure might be related to seasonal variations in the Brewer–Dobson circulation.

The third interesting feature is hemispheric asymmetry of summer-time ozone trend patterns below 35 km: they are negative in the NH and positive in the SH. No similar analyses exist for temperature, and this can be the subject of future work. Hemispheric asymmetry was reported in recent studies (Froidevaux et al., 2019, Ball et al., 2019), even if not broken down by season. Froidevaux et al. (2019) showed that ozone trends derived from Aura Microwave Limb Sounder (Aura/MLS) data over a shorter period (2005–2018) have a tendency towards slightly positive values in the SH. Additionally, this asymmetry
might be related to the hydrogen chloride (HCl) abundances and trends (Mahieu et al., 2014, Han et al., 2019). At the moment, we can only speculate that this might also be a contributing factor to the observed ozone negative trends in that region.



## 6 Summary

Using four long-term merged data sets of ozone profiles we have studied the seasonal dependence of ozone trends in the stratosphere. The results of our analysis, based on two-step multiple linear regression can be summarized as follows:

- The upper stratospheric ozone is recovering, and the recovery maximizes during local winters and equinoxes reaching up to 3-4% dec$^{-1}$, which is fully consistent with weaker upper-stratospheric cooling in local winter at mid- and high latitudes.
- In the tropics, there is very strong seasonal dependence of ozone trends at all altitudes. The trends are changing from positive to negative, with the sign of transition depending on altitude and season.
- Below 25 km in the tropical region, strong negative trends are observed during spring and summer, which are statistically significant for all datasets and consistent with the seasonal pattern of temperature trends in this region.
- In the lower and middle stratosphere, there is hemispheric asymmetry during the local summers and equinoxes at mid-latitudes with negative trend in the North and positive trend in the South.

Despite some discrepancies, the general coherence in trends derived from four different merged data sets gives us confidence in the validity and robustness of the results. We compared the seasonal dependence of ozone trends with available analyses of the seasonal dependence of stratospheric temperature trends and found a clear inter-relation of the trend patterns.





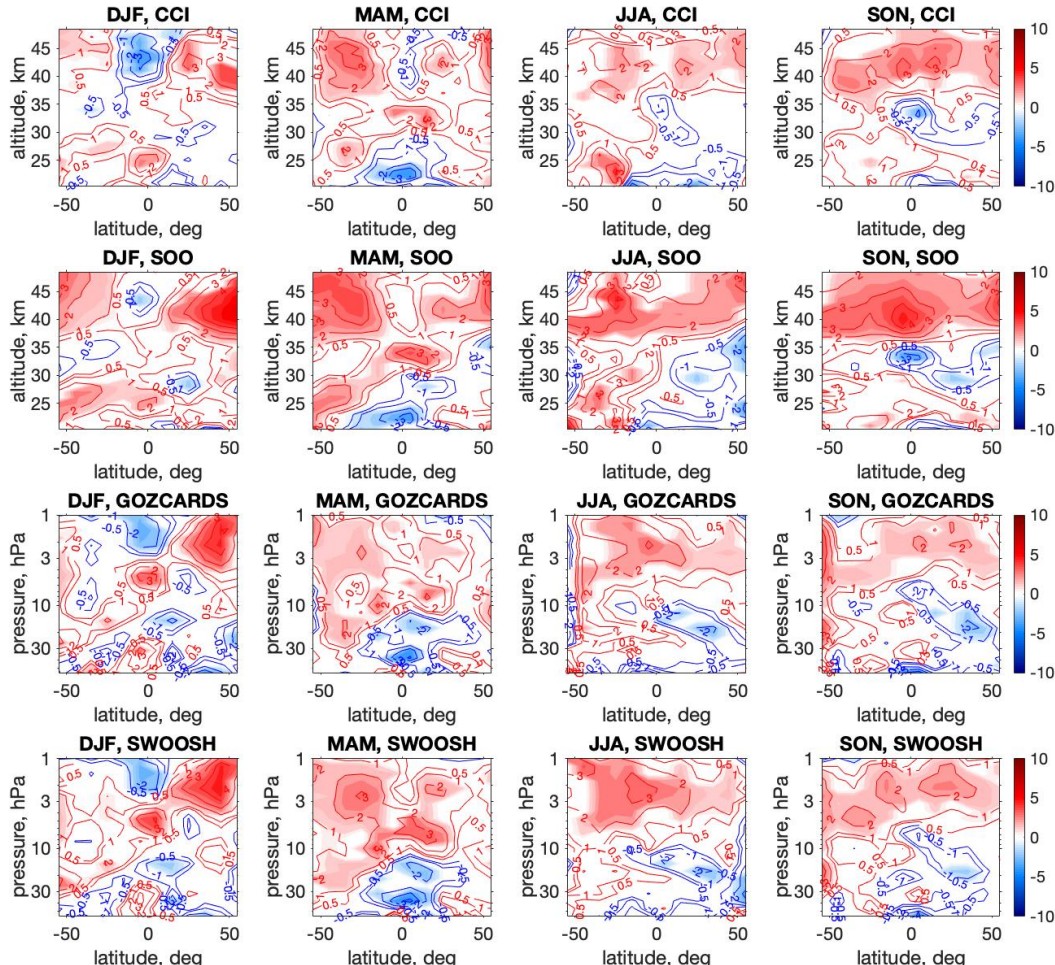

**Figure 6. Altitude-latitude variation of linear trends in ozone calculated over 2000-2018 for each season. Data are presented on their natural vertical coordinate: altitude grid for CCI and SOO and pressure grid for GOZCARDS and SWOOSH. The shading denotes trends that are significant at the 95% level.**

**Data availability**

The SAGE-CCI-OMPS dataset is available from CCI website (http://www.esa-ozone-cci.org). The SAGEII-OSIRIS-OMPS dataset is available from University of Saskatchewan ftp-site. GOZCARDS ozone data updates (version 2.20) are available by contacting Lucien Froidevaux; this data version will also be updated on the public Goddard Earth Sciences Data and

10    Information Services Center (GES DISC) website (https://disc.gsfc.nasa.gov) in the near future.



## Acknowledgements

The SAGE-CCI-OMPS dataset was created within the ESA Ozone_CCI project. V.S and M.E.S thank the Academy of Finland (Projects INQUIRE and SECTIC, Center of Excellence of Inverse Modelling and Imaging). The authors thank the Canadian Space Agency. Work at the Jet Propulsion Laboratory, California Institute of Technology, was performed under contract with
the National Aeronautics and Space Administration (NASA). We acknowledge the essential contributions from R. Wang, J. Anderson, and R. Fuller to the GOZCARDS data records used here.

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
