# Peer review of "Seasonal stratospheric ozone trends over 2000-2018 derived from several merged data sets."

_Atmospheric Chemistry and Physics, 2019_

## Referee Comment (RC1) · Anonymous Referee #1 · 4 Feb 2020

This work uses four composite data sets used in the last ozone assessment as part of a regression analysis to determine the seasonal trends in stratospheric ozone. The methodology is similar to what has been used in previous works and is simple and straightforward. However, applying this methodology to what amounts to much smaller data sets requires some modifications to avoid very large uncertainties. These modifications require some caveats to be stated to avoid potential misunderstanding the limitations of the significance of the results. Otherwise, the results are interesting and appear mostly consistent with previously published work on temperature trends in the stratosphere, though there are some discrepancies with other published work that need to be addressed to ensure accuracy of the work performed here.

[Figure]

**Major Comments**

Pg. 04, Ln. 10: "The two-step approach is equivalent to the one-step regression . . ."
The two approaches are not equivalent. While the results of the fitting coefficients tend to be very close to the same (it depends on the nature of the data and the regression model), the truth is that the uncertainty analysis becomes less robust, because covariance information between dependence on the different proxies is lost with each step. The multistep procedure tends to result in smaller uncertainties than a single step procedure, but this is because those uncertainties are artificially biased low from the lost information, not because it is necessarily a better process. On that note, technically this is a three-step regression since it is performed on deseasonalized anomalies (i.e., the first step). In lieu of doing things different, I would merely state that the multi-step approach is meant to avoid the use of more complicated regression models applied to a reduced amount of data at the expense of potentially biasing the uncertainties low.

The authors describe their "two-step approach" on page 4 (Lns. 15–33) where the data is fit with Eqn. 1 with more data to derive the natural cycles before removing them and then fitting two different trends over two different time periods. This is intended to offer sufficient data to properly fit the natural variability. Generally the wording here is fine (e.g., "thus providing more accurate fitting of these proxies" and "thus providing smaller sensitivity to defining turnaround point") but I would add a caveat. Namely, Eqn 1 allows for the use of the PWLT term to potentially alias into some of this natural variability such that its removal for step 2 will affect the residuals when the trends are fit.

Pg. 05, Ln. 15: The paragraph that starts here (and goes onto the next page) discusses potential reasons why Method 1 would result in smaller uncertainties than Method 2. The seasonal dependence of the QBO, for example, is a likely culprit but multiple

potential reasons exist. One is simply that using data from all seasons would require a more complicated QBO model in the regression and so the residuals are greater and potentially seasonally-dependent. This is easily analyzed by looking at the amplitude of the QBO in the regression from the coefficients between the different Methods. However, another is that the uncertainties in the trends computed in step 2 are done without any consideration of the uncertainties in the residuals from which they were fit. This could be artificially biasing those trend uncertainties lower in Method 1, which ties back into my first Major Comment. I would worry about what the influence of a few potential outliers are on the trends without any of the covariance information between proxies getting captured in the uncertainty analysis. In general I do not argue with the notion of picking a Method and sticking with it, simply that the caveats about the different methods need to be mentioned/discussed in the paper. This methodology has its uses, but it has its limitations as well.

Pg. 10, Lns. 35–41 discuss the hemispheric asymmetry of the summer trends in the middle stratosphere at mid-latitudes (25-35km). This made me think to compare this work with what was shown in the last ozone assessment, as both use the same data sets. While WMO (2018) does not attempt to look at the seasonal trends, it does look at the overall trends. As such, I would expect the black lines in Figs. 5 and S3 to agree with what is shown in Fig. 3-19 of the Assessment. While much of the results are in agreement, I see some clear offsets and discrepancies, particularly between CCI in the Southern/Northern Hemisphere and SOO in the Northern Hemisphere. This makes me wonder if the term "All" in the figures in this paper are using all of the data at once in the analysis or if they are instead some sort of average of the results from different seasons. If the former, I would expect better agreement between this work and the Assessment. If the latter, then each value should be weighted by its seasonal mean to better represent the former, otherwise it does not really carry any meaning. This does not necessarily mean that the results from the seasonal regressions are incorrect, but it does make me wonder if double checking that all of the results shown

here are accurate is needed. If the data presented are accurate, then I would want to know why they disagree with those results from the Assessment.

**Minor Comments**

Pg. 01, Ln. 35: "The Antarctic ozone hole is showing some signs of recovery, and the first signatures of global recovery . . ."
I would recommend adding a reference to the first part of this sentence such as Solomon et al. (2016) (DOI: 10.1126/science.aae0061)

Pg. 04, Ln. 01: "The CCI and SOO datasets provide deseasonalized ozone anomalies. For GOZCARDS and SWOOSH, the deseasonalized anomalies were computed in the same way. The seasonal cycle was evaluated using the data from 2005-2011."
Was the seasonal cycle computed as averages of data in each month or using some sinusoidal fit? I am assuming the former but please specify.

Pg. 04, Ln. 25: ". . . 2-month lagged ENSO proxy . . ."
Why two months?

Figs. 3 and 6 are very busy (and large) figures and have the same information content. Wouldn't it make more sense to only include one of them in the paper for discussion? The ratio of figure space to text space is quite large.

---

## Referee Comment (RC2) · Anonymous Referee #2 · 4 Mar 2020

Review of "Seasonal stratospheric ozone trends over 2000-2018 derived from several merged data sets" by M.E. Szelag et al.

This manuscript describes an analysis of seasonally-varying long-term trends in stratospheric ozone, focusing on the period from 2000-2018. The authors analyze and compare data from four high vertical resolution merged ozone data sets. The analysis follows well-established procedures using multiple linear regression to isolate long-term trend from other sources of variability. The paper is well organized and concise. The authors discuss the consistency of the ozone results with previously published seasonally-dependent stratospheric temperature trends. The temperature and ozone trends have comparable seasonal characteristics, which the authors reasonably suggest might be due to seasonal changes in the Brewer Dobson circulation in time (with

changing climate), such that in the upper stratosphere the mean ozone trend (positive) is due to photochemistry and decreasing ODS's, while the seasonal variability is dynamical. Highlighting the seasonal component of the trends in profile ozone is an important addition and is appropriate for publication in ACP. I recommend publication after the following comments, mostly minor, are addressed.

Of the four data sets, two have native units of volume mixing ratio on pressure, and two have native units of number density on altitude. If I understand correctly, each data set is analyzed and plotted in its native units. Could the authors say more about whether these should be directly comparable? They would not be if the pressure surfaces are changing in time relative to the altitude surfaces, that is, in the presence of a temperature trend. (see McLinden and Fioletov, https://agupubs.onlinelibrary.wiley.com/doi/full/10.1029/2010GL046012). I believe the answer is the temperature trends over the period are small, but this should be stated. Also, the seasonally-varying temperature trends mean this effect will differ by season. Do the authors see differences in the number density vs VMR trends that are consistent with seasonal temperature trends?

In the tropics, the authors show that both the trend and uncertainty can vary substantially between their method 1 (use seasonal time series only to fit and remove natural variability) and their method 2 (use full time series to remove variability). The authors state that fitting the natural variability proxies with insufficient number of points will cause larger uncertainties (P4 L17), and in P4 L30 that the two-step approach provides for a sufficient number of points. However, is this true for Method 1? In Method 1, only seasonal points are fit, and although three months of each year are fit, the proxies are all slowly varying such that they are highly correlated over the three-month segments of each year, so I question the assumption that there is a sufficient number of points to accurately fit to the proxies. The authors find that the uncertainty is generally less using method 1, but I'm not sure this is a true reduction in uncertainty. With the smaller number of points being fit, there may be higher correlation between proxies that

is arbitrary (i.e. not physical). This would allow more cross-talk between the proxies in the regression, and thus a better fit, but not a physically meaningful fit. For example if there was correlation between the seasonal QBO and PWLT, the best fit in the regression might be a high coefficient on the QBO term, and an equally high but opposite sign coefficient on the PWLT, which when added allow the regression to better fit some short-term variability while the large-scale changes cancel. This would lead to smaller short-term variability in the residual (after QBO fit removed), but a larger PWLT term left behind, which then affects the trend segment fits in step 2. This is not necessarily the case, and the correspondence with the temperature trends lends support to the results being physical, but these caveats should be discussed. It would be ok to refer this to future work, but to investigate these possibilities, the authors might compare the derived ozone variability based on the full QBO proxy to that from the seasonal QBO proxies. Are the seasonal variations consistent with how we expect the ozone QBO to behave seasonally, and are they consistent in latitude and altitude, or generally noisy?

Minor Comments: P1 L36 Is there a specific reference for the Antarctic ozone hole recovery (more recent than 2015) in addition to WMO 2018?

P3 Table 1 Under Ozone Profile Representation, are the first two entries (deseasonalized anomalies) in units of number density (or concentration) or percent? I suggest adding the units to each column.

P4 L2-3 Suggest "For GOZCARDS and SWOOSH, the deseasonalized anomalies were computed relative to their 2005-2011 mean seasonal cycle." [These were not necessarily computed the same way as those initially provided as anomalies. I suspect that for those, the seasonal cycle was subtracted from each instrument individually before merging, which, if there are seasonal biases in the individual records, is different than using the seasonal cycle of the final merged record.]

P4 L9-10 Suggest re-wording slightly, "The two-step approach allows us to avoid fitting over the period when the ozone trends transition from negative to positive, and are not

well-represented by a linear function." I want to get across the idea that it is not only that we don't know the exact turn-around time (and this varies with latitude and altitude) but that the ozone change is not linear over this period anyway, so there may not be a well-defined turn-around time.

P4 L12-14 Suggest dropping the last sentence, as using dynamical linear modelling is not otherwise discussed.

P4 L27 certain period -> certain season

P4 L31-33 Similar to above, suggest slight re-wording of the last sentence. The authors do fit near the "trend turnaround point in 1997" if the decline segment fit is 1984-1997. It is not so much sensitivity to the turnaround time as it is that the ozone change in time does not look like the hockey stick representation (it is a curve rather than two intersecting slopes). What about "In the second step, fitting is only done during periods when the ozone change is approximately linear, thus avoiding the problem of how to properly model the ozone change in the trend turnaround period (such as sensitivity to trend turnaround time when using a hockey-stick representation)."

P5 L15 Related to above comments, with fewer points, the individual proxies may be more correlated, giving the regression more room to "play" (more degrees of freedom) and get a better fit, but that fit might not be physical.

P8 L1 Just clarifying, the analysis is redone for ozone averaged in the broad latitude bands, as opposed to averaging the trends in the smaller bands. This is how I read the text, I just want to be sure.

Editorial Comments/Typos: P1 L13, add comma after SWOOSH

P2 L11 -1 K dec-1 (add space)

P3 Table 1 Suggest some re-wording on the Merging Method: Median value of deseasonalized anomalies Average value of deseasonalized anomalies referenced to SAGE II Average value of original values referenced to SAGE II Average value of original

values referenced to SAGE II

P4 L4 Trend analyses are usually performed using a multiple linear regression...

P4 L8 ... two steps, by detecting and removing natural cycles in the first step, and estimating bulk changes over specific periods in the second step ...

P4 L30 In the two-step approach, *a* sufficient number ...

P4 L 34 Figure 2 illustrates each step of our analysis ...

P4 L35 Seasonal data (3-months, method #1)

P5 L1 estimated for the period 2000-2008

P5 L9 like in -> as in

P6 Figure 2 Caption Last sentence, Right panel shows...

P6 L16 Figure 2 -> Figure 3

P7 Figure 3 GOZCARDS and SWOOSH titles give altitude range, but shouldn't these give pressure ranges. The caption also says pressure bands. I would suggest changing the word bands to ranges in the caption, but also include the pressure range in the plot titles rather than the altitude range.

P9 L8 are dominating over -> dominate in

P10 L2 Figure S4 does not match Figure 6.

P10 L10 larger positive ozone trends

P10 L14 due to greenhouse gas (remove 'the')

P10 L30 so do -> as are

P10 L31 additional confirmation of our hypothesis

---

## Author Comment (AC1) · 16 Apr 2020

We would like to thank Reviewer #1 for the comments and the effort that the reviewer has put into this paper. Please see our response below.

MAJOR COMMENTS

Reviewer: Pg. 04, Ln. 10: "The two-step approach is equivalent to the one-step regression . . ." The two approaches are not equivalent. While the results of the fitting coefficients tend to be very close to the same (it depends on the nature of the data and the regression model), the truth is that the uncertainty analysis becomes less robust, because covariance information between dependence on the different proxies is lost with each step. The multistep procedure tends to result in smaller uncertainties than

a single step procedure, but this is because those uncertainties are artificially biased low from the lost information, not because it is necessarily a better process. On that note, technically this is a three-step regression since it is performed on deseasonalized anomalies (i.e., the first step). In lieu of doing things different, I would merely state that the multi-step approach is meant to avoid the use of more complicated regression models applied to a reduced amount of data at the expense of potentially biasing the uncertainties low.

Authors: In the revised version, we will write "nearly equivalent". We would like to note that the autocorrelations of residuals are taken into account in our regression method also in the second step, therefore the method should not bias the uncertainties low. In the revised version, we will add that autocorrelations are taken into account in both steps.

Reviewer: The authors describe their "two-step approach" on page 4 (Lns. 15–33) where the data is fit with Eqn. 1 with more data to derive the natural cycles before removing them and then fitting two different trends over two different time periods. This is intended to offer sufficient data to properly fit the natural variability. Generally, the wording here is fine (e.g., "thus providing more accurate fitting of these proxies" and "thus providing smaller sensitivity to defining turnaround point") but I would add a caveat. Namely, Eqn 1 allows for the use of the PWLT term to potentially alias into some of this natural variability such that its removal for step 2 will affect the residuals when the trends are fit.

Authors: Of course, any method for detecting cycles includes assumptions about the trend function (linear, piecewise, smooth), which might affect the estimated amplitude of the cycles. This is not a caveat for our method, but for multiple linear regression in general.

Reviewer: Pg. 05, Ln. 15: The paragraph that starts here (and goes onto the next page) discusses potential reasons why Method 1 would result in smaller uncertainties

than Method 2. The seasonal dependence of the QBO, for example, is a likely culprit but multiple potential reasons exist. One is simply that using data from all seasons would require a more complicated QBO model in the regression and so the residuals are greater and potentially seasonally dependent. This is easily analyzed by looking at the amplitude of the QBO in the regression from the coefficients between the different Methods.

Authors: Indeed, the QBO amplitude is different in different seasons.

Reviewer: However, another is that the uncertainties in the trends computed in step 2 are done without any consideration of the uncertainties in the residuals from which they were fit.

Authors: No, uncertainties at the 2nd step are estimated from the fit residuals. We will state this in the revised version (PG. 05, Ln. 35). The uncertainties are smaller in method #1 because of better fitting the cycles (smaller residuals).

Reviewer: Pg. 10, Lns. 35–41 discuss the hemispheric asymmetry of the summer trends in the middle stratosphere at mid-latitudes (25-35km). This made me think to compare this work with what was shown in the last ozone assessment, as both use the same data sets. While WMO (2018) does not attempt to look at the seasonal trends, it does look at the overall trends. As such, I would expect the black lines in Figs. 5 and S3 to agree with what is shown in Fig. 3-19 of the Assessment. While much of the results are in agreement, I see some clear offsets and discrepancies, particularly between CCI in the Southern/Northern Hemisphere and SOO in the Northern Hemisphere. This makes me wonder if the term "All" in the figures in this paper are using all of the data at once in the analysis or if they are instead some sort of average of the results from different seasons. If the former, I would expect better agreement between this work and the Assessment. If the latter, then each value should be weighted by its seasonal mean to better represent the former, otherwise it does not really carry any meaning. This does not necessarily mean that the results from the seasonal regressions are

incorrect, but it does make me wonder if double checking that all of the results shown here are accurate is needed. If the data presented are accurate, then I would want to know why they disagree with those results from the Assessment.

Authors: In our analysis, yearly trend estimates ("all") are obtained using all the data at once. We would like to note that:

(1) slightly different latitude zones are used in the WMO (2018) report;

(2) different time periods are used in WMO assessment (2000-2016) and in our paper (2000-2018);

(3) different regression methods are used in WMO-2018 report (independent linear trend regression) and in our work (two-step regression).

And, despite these differences, the trends in WMO-2018 report and yearly trends in our paper are close to each other. We compared directly the trends presented in Figure 3-19 of WMO-2018 report (V. Sofieva produced this figure) with the yearly trends from our analyses, they are shown in Figures 1 and Figure 2 below.

The most pronounced difference is the reduced trends in Northern Hemisphere. Most probably, the main impact is due to two-year extension. This reduction of trends due to 2 years extension is also observed using traditional trend analysis.

MINOR COMMENTS

Reviewer: Pg. 01, Ln. 35: "The Antarctic ozone hole is showing some signs of recovery, and the first signatures of global recovery . . ." I would recommend adding a reference to the first part of this sentence such as Solomon et al. (2016) (DOI: 10.1126/science.aae0061)

Authors: We now add the reference as suggested by the Reviewer.

Reviewer: Pg. 04, Ln. 01: "The CCI and SOO datasets provide deseasonalized ozone anomalies. For GOZCARDS and SWOOSH, the deseasonalized anomalies were computed in the same way. The seasonal cycle was evaluated using the data from 2005-2011." Was the seasonal cycle computed as averages of data in each month or using some sinusoidal fit? I am assuming the former but please specify.

Authors: Yes, the seasonal cycle is computed via averaging the data in each month. We will clarify this in the revised version.

Reviewer: Pg. 04, Ln. 25: ". . . 2-month lagged ENSO proxy . . ." Why two months?

Authors: The MEI ENSO index is defined as a 2-month average. It was used with 2-month lag in several trend analyses, for example, (Bourassa et al., 2014; Randel and Thompson, 2011; Sofieva et al., 2017). The reason for the lagging is somewhat smaller residuals in the stratosphere. However, the sensitivity of the fit to lagging by 1-2 months is small (Petropavlovskikh et al., 2019). In the revised version, we clarify this.

Reviewer: Figs. 3 and 6 are very busy (and large) figures and have the same information content. Wouldn't it make more sense to only include one of them in the paper for discussion? The ratio of figure space to text space is quite large.

Authors: We think that Figure 6 is useful for discussion, and we would prefer to keep it in the manuscript.

References

Bourassa, A. E., Degenstein, D. A., Randel, W. J., Zawodny, J. M., Kyrölä, E., McLinden, C. A., Sioris, C. E. and Roth, C. Z.: Trends in stratospheric ozone derived from merged SAGE II and Odin-OSIRIS satellite observations, Atmos. Chem. Phys., 14(13), 6983–6994, doi:10.5194/acp-14-6983-2014, 2014. Randel, W. J. and Thompson, A. M.: Interannual variability and trends in tropical ozone derived from SAGE II satellite data and SHADOZ ozonesondes, J. Geophys. Res. Atmos., 116(D07303), doi:10.1029/2010JD015195, 2011. Sofieva, V. F., Kyrölä, E., Laine, M., Tamminen, J., Degenstein, D., Bourassa, A., Roth, C., Zawada, D., Weber, M., Rozanov, A., Rahpoe, N., Stiller, G., Laeng, A., von Clarmann, T., Walker, K. A., Sheese, P., Hubert, D., van Roozendael, M., Zehner, C., Damadeo, R., Zawodny, J., Kramarova, N. and Bhartia, P. K.: Merged SAGE II, Ozone_cci and OMPS ozone profile dataset and evaluation of ozone trends in the stratosphere, Atmos. Chem. Phys., 17(20), 12533–12552, doi:10.5194/acp-17-12533-2017, 2017. Petropavlovskikh, I., Godin-Beekmann, S., Hubert, D., Damadeo, R., Hassler, B. and Sofieva, V.: SPARC/IO3C/GAW Report on Long-term Ozone Trends and Uncertainties in the Stratosphere, edited by M. Kenntner and B. Ziegele, [online] Available from: https://elib.dlr.de/126666/, 2019.

[Figure]

**Fig. 1.** Comparison of post-2000 ozone trends for SAGE-CCI-OMPS datasets. Cyan lines: WMO 2018 assessment (2000-2016), black lines: yearly trends in our work (years 2000-2018).

[Figure]

**Fig. 2.** As Figure 1, but for SAGE-OSIRIS-OMPS dataset (SOO). Blue lines: trends from WMO-2018 assessment, black line: our analysis.

---

## Author Comment (AC2) · 16 Apr 2020

We would like to thank Reviewer #2 for the comments and the effort that the reviewer has put into this paper. Please see our response below.

**MAJOR COMMENTS**

Reviewer: Of the four data sets, two have native units of volume mixing ratio on pressure, and two have native units of number density on altitude. If I understand correctly, each data set is analyzed and plotted in its native units. Could the authors say more about whether these should be directly comparable? They would not be if the pressure surfaces are changing in time relative to the altitude surfaces, that is, in the presence of a temperature trend. (see McLinden and Fioletov, doi: 10.1029/2010GL046012). I

believe the answer is the temperature trends over the period are small, but this should be stated. Also, the seasonally-varying temperature trends mean this effect will differ by season. Do the authors see differences in the number density vs VMR trends that are consistent with seasonal temperature trends?

Authors: We agree, the ozone trends in different representations and vertical coordinate can be different due to temperature trends. However, since the temperature trends after 2000 are small, we expect also a minor difference in ozone trends in different ozone representations. In the revised version, we note this and add the reference to the paper by McLinden and Fioletov, 2011. We have plot the difference of seasonal trends for mean(CCI, SOO) minus mean(GOZCARDS, SWOOSH), similar way to Figure 4 in the manuscript. The difference in the upper stratosphere is consistent with the pattern on temperature trends and predictions by McLinden and Fioletov (see Figure 1 below). In the revised version, we include this figure into the Supplement.

Reviewer: In the tropics, the authors show that both the trend and uncertainty can vary substantially between their method 1 (use seasonal time series only to fit and remove natural variability) and their method 2 (use full time series to remove variability). The authors state that fitting the natural variability proxies with insufficient number of points will cause larger uncertainties (P4 L17), and in P4 L30 that the two-step approach provides for a sufficient number of points. However, is this true for Method 1? In Method 1, only seasonal points are fit, and although three months of each year are fit, the proxies are all slowly varying such that they are highly correlated over the three-month segments of each year, so I question the assumption that there is a sufficient number of points to accurately fit to the proxies. The authors find that the uncertainty. With the smaller number of points being fit, there may be higher correlation between proxies that is arbitrary (i.e. not physical). This would allow more cross-talk between the proxies in the regression, and thus a better fit, but not a physically meaningful fit. For example if there was correlation between the seasonal QBO and PWLT, the best fit
in the regression might be a high coefficient on the QBO term, and an equally high but opposite sign coefficient on the PWLT, which when added allow the regression to better fit some short-term variability while the large-scale changes cancel. This would lead to smaller short-term variability in the residual (after QBO fit removed), but a larger PWLT term left behind, which then affects the trend segment fits in step 2. This is not necessarily the case, and the correspondence with the temperature trends lends support to the results being physical, but these caveats should be discussed. It would be ok to refer this to future work, but to investigate these possibilities, the authors might compare the derived ozone variability based on the full QBO proxy to that from the seasonal QBO proxies. Are the seasonal variations consistent with how we expect the ozone QBO to behave seasonally, and are they consistent in latitude and altitude, or generally noisy?

Authors: In the revised version, we add caveats that the correlation between proxies can be different for Method 1 and Method 2, thus the reduction of uncertainty can be not fully physical. The detailed analyses of proxy correlations can be subject of future studies.

**MINOR COMMENTS**

Reviewer: P1 L36 Is there a specific reference for the Antarctic ozone hole recovery (more recent than 2015) in addition to WMO 2018?

Authors: We added the reference (Solomon et al., 2016).

Reviewer: P3 Table 1 Under Ozone Profile Representation, are the first two entries (deseason- alized anomalies) in units of number density (or concentration) or percent? I suggest adding the units to each column.

Authors: We now add the units to each column in Table 1.

Reviewer: P4 L2-3 Suggest "For GOZCARDS and SWOOSH, the deseasonalized anomalies were computed relative to their 2005-2011 mean seasonal cycle." [These
were not necessarily computed the same way as those initially provided as anomalies. I suspect that for those, the seasonal cycle was subtracted from each instrument individually be- fore merging, which, if there are seasonal biases in the individual records, is different than using the seasonal cycle of the final merged record.]

Authors: Corrected as suggested by the Reviewer.

Reviewer: P4 L9-10 Suggest re-wording slightly, "The two-step approach allows us to avoid fitting over the period when the ozone trends transition from negative to positive and are not well-represented by a linear function." I want to get across the idea that it is not only that we don't know the exact turn-around time (and this varies with latitude and altitude) but that the ozone change is not linear over this period anyway, so there may not be a well-defined turn-around time.

Authors: Corrected as suggested by the Reviewer.

Reviewer: P4 L12-14 Suggest dropping the last sentence, as using dynamical linear modelling is not otherwise discussed.

Authors: Corrected as suggested by the Reviewer.

Reviewer: P4 L27 certain period -> certain season

Authors: Corrected as suggested by the Reviewer.

Reviewer: P4 L31-33 Similar to above, suggest slight re-wording of the last sentence. The authors do fit near the "trend turnaround point in 1997" if the decline segment fit is 1984-1997. It is not so much sensitivity to the turnaround time as it is that the ozone change in time does not look like the hockey stick representation (it is a curve rather than two intersecting slopes). What about "In the second step, fitting is only done during periods when the ozone change is approximately linear, thus avoiding the problem of how to properly model the ozone change in the trend turnaround period (such as sensitivity to trend turnaround time when using a hockey-stick representation)."
Authors: Corrected as suggested by the Reviewer.

Reviewer: P5 L15 Related to above comments, with fewer points, the individual proxies may be more correlated, giving the regression more room to "play" (more degrees of freedom) and get a better fit, but that fit might not be physical.

Authors: Please see our replies above, in the major comments.

Reviewer: P8 L1 Just clarifying, the analysis is redone for ozone averaged in the broad latitude bands, as opposed to averaging the trends in the smaller bands. This is how I read the text, I just want to be sure.

Authors: That is correct.

EDITORIAL COMMENTS/TYPOS

Reviewer: P1 L13, add comma after SWOOSH

Authors: Corrected as suggested by the Reviewer.

Reviewer: P2 L11 -1 K dec-1 (add space)

Authors: Corrected as suggested by the Reviewer.

Reviewer: P3 Table 1 Suggest some re-wording on the Merging Method: Median value of desea- sonalized anomalies Average value of deseasonalized anomalies referenced to SAGE II Average value of original values referenced to SAGE II Average value of original values referenced to SAGE II Average value of original values referenced to SAGE II

Authors: Corrected as suggested by the Reviewer.

Reviewer: P4 L4 Trend analyses are usually performed using a multiple linear regression. . .

Authors: Corrected as suggested by the Reviewer.

Reviewer: P4 L8 . . . two steps, by detecting and removing natural cycles in the first
step, and estimating bulk changes over specific periods in the second step . . . Authors: Corrected as suggested by the Reviewer. Reviewer: P4 L30 In the two-step approach, \*a\* sufficient number . . . Authors: Corrected as suggested by the Reviewer. Reviewer: P4 L 34 Figure 2 illustrates each step of our analysis . . . Authors: Corrected as suggested by the Reviewer. Reviewer: P4 L35 Seasonal data (3-months, method #1) Authors: Corrected as suggested by the Reviewer. Reviewer: P5 L1 estimated for the period 2000-2008 Authors: Corrected as suggested by the Reviewer. Reviewer: P5 L9 like in -> as in Authors: Corrected as suggested by the Reviewer. Reviewer: P6 Figure 2 Caption Last sentence, Right panel shows. . . Authors: Corrected as suggested by the Reviewer. Reviewer: P6 L16 Figure 2 -> Figure 3

Authors: Corrected as suggested by the Reviewer.

Reviewer: P7 Figure 3 GOZCARDS and SWOOSH titles give altitude range, but shouldn't these give pressure ranges. The caption also says pressure bands. I would suggest changing the word bands to ranges in the caption, but also include the pressure range in the plot titles rather than the altitude range.

Authors: Corrected as suggested by the Reviewer.
Reviewer: P9 L8 are dominating over -> dominate in

Authors: Corrected as suggested by the Reviewer.

Reviewer: P10 L2 Figure S4 does not match Figure 6.

Authors: The part with Supplementary Figure S4 is now moved to correct place in the manuscript (Description of the Figure 4). Note, that the order of Supplementary Figures has changed.

Reviewer: P10 L10 larger positive ozone trends

Authors: Corrected as suggested by the Reviewer.

Reviewer: P10 L14 due to greenhouse gas (remove 'the')

Authors: Corrected as suggested by the Reviewer.

Reviewer: P10 L30 so do -> as are

Authors: Corrected as suggested by the Reviewer.

Reviewer: P10 L31 additional confirmation of our hypothesis

Authors: Corrected as suggested by the Reviewer.
**Fig. 1.** Altitude-seasons variation of the difference of the seasonal trends (mean[CCI, SOO] minus mean[GOZCARDS, SWOOSH]), calculated over 2000-2018 for three selected latitudinal bands.